# Protocol for an 'efficient design' cluster randomised controlled trial to evaluate a complex intervention to improve antibiotic prescribing for CHIldren presenting to primary care with acute COugh and respiratory tract infection: the CHICO study

Penny Seume,[1] Scott Bevan,[1] Grace Young,[2] Jenny Ingram [ID],[3] Clare Clement [ID],[2] Christie Cabral,[1] Patricia Jane Lucas,[4] Elizabeth Beech,[5] Jodi Taylor,[2] Jeremy Horwood,[1] Padraig Dixon,[1] Martin C Gulliford [ID],[6] Nick Francis [ID],[7] Sam T Creavin,[1] Athene Lane,[2] Alastair D Hay,[1] Peter S Blair [ID] [3]

For numbered affiliations see end of article.

**Correspondence to**
Professor Peter S Blair;
p.s.blair@bris.ac.uk

## ABSTRACT

**Introduction** Respiratory tract infections (RTIs) in children are common and present major resource implications for primary care. Unnecessary use of antibiotics is associated with the development and proliferation of antimicrobial resistance. In 2016, the National Institute for Health Research (NIHR)-funded 'TARGET' programme developed a prognostic algorithm to identify children with acute cough and RTI at very low risk of 30-day hospitalisation and unlikely to need antibiotics. The intervention includes: (1) explicit elicitation of parental concerns, (2) the results of the prognostic algorithm accompanied by prescribing guidance and (3) provision of a printout for carers including safety netting advice. The CHIldren's COugh feasibility study suggested differential recruitment of healthier patients in control practices. This phase III 'efficiently designed' trial uses routinely collected data at the practice level, thus avoiding individual patient consent. The aim is to assess whether embedding a multifaceted intervention into general practitioner (GP) practice Information Technology (IT) systems will result in reductions of antibiotic prescribing without impacting on hospital attendance for RTI.

**Methods and analysis** The coprimary outcomes are (1) practice rate of dispensed amoxicillin and macrolide antibiotics, (2) hospital admission rate for RTI using routinely collected data by Clinical Commissioning Groups (CCGs). Data will be collected for children aged 0–9 years registered at 310 practices (155 intervention, 155 usual care) over a 12-month period. Recruitment and randomisation of practices (using the Egton Medical Information Systems web data management system) is conducted via each CCG stratified for children registered and baseline dispensing rates of each practice. Secondary outcomes will explore intervention effect modifiers. Qualitative interviews will explore intervention usage. The

### Strengths and limitations of this study

► Informed by a feasibility study this 'efficient-design' cluster randomised controlled trial uses routinely collected aggregated measures for the coprimary outcomes, and avoids postrandomisation recruitment bias associated with individual patient consent.

► The study will recruit practices across England, thus including research-naïve practices and those serving diverse socioeconomic populations.

► The complex intervention, embedded within practice electronic health records, stems from a 5-year NIHR-funded programme and includes: (1) a prognostic algorithm to stratify children's risk of hospitalisation due to respiratory infection in the following 30 days; (2) tools to improve patient-doctor communication; and (3) home care information (an alternate treatment action for clinicians).

► The design only allows for dispensing to be related to the number of children registered at the practice rather than the number consulting for respiratory tract infections (RTI), and it will not allow quantification of delayed prescribing.

► The other primary outcome is hospitalisation for RTI, and this relies on the quality of the data collected by Clinical Commissioning Groups—any difficulties obtaining this information or limitations of this efficient design will be reported.

economic evaluation will be limited to a between-arm comparison in a cost–consequence analysis.

**Ethics and dissemination** Research ethics approval was given by London-Camden and Kings Cross Research Ethics Committee (ref:18/LO/0345). This manuscript refers

to protocol V.4.0. Results will be disseminated through peer-reviewed journals and international conferences.

**Trial registration number** ISRCTN11405239.

## INTRODUCTION
### Background
Acute respiratory tract infections (RTI) in children are a common reason for antibiotic prescribing. In English primary care, most antibiotics are prescribed for conditions that only sometimes require antibiotic treatment, depending on patient-specific indicators.[1] Although there has been a decline in prescribing for uncomplicated RTI in England over the last decade, more than a third of children were still prescribed antibiotics for these infections.[2] Clinical uncertainty in primary care regarding the prognosis of children with RTIs (ie, knowing which children will and won't subsequently deteriorate) contributes to the unnecessary use of existing antibiotics, which is associated with increasing antimicrobial resistance.[3 4] Qualitative work from our 5-year National Institute for Health Research (NIHR)-funded 'TARGET' programme grant, completed in 2016, identified this uncertainty as a major driver of antibiotic prescribing.[5] We hypothesised that improved identification of children at very low risk of future hospitalisation might help reduce clinical uncertainty.[6] As part of the 'TARGET' programme, we developed a prognostic algorithm that could be used by clinicians to identify children at very low risk of hospitalisation as well as tools to improve patient—doctor communication.[7]

### Lessons learnt from the feasibility cluster randomised controlled trial
Findings from across the 'TARGET' programme were used to develop a complex intervention designed to reduce antibiotic prescribing. The subsequent feasibility cluster randomised controlled trial (RCT) for CHIldren's COugh (CHICO) showed significant prescribing reductions in both arms of the trial compared with the cohort data of the programme but also exposed both lower prescribing levels and differential recruitment of healthier children in the control arm.[8] In the qualitative interviews, clinicians reported preferential recruitment of less unwell children as these were quicker to manage and therefore easier to recruit. To negate differential recruitment, and conserve resources, an 'efficient design' was proposed for the full trial. Efficient design trials often use routinely collected data.[9] In the case of CHICO using aggregated data, this both avoids the need for individual patient consent (and differential recruitment) and utilises existing practice level data. This simpler design, placing fewer demands on clinicians and practices compared with other studies, will also encourage the recruitment of research-naïve practices. The primary outcomes are routinely collected antibiotic dispensing data, collected by ePACT2 for the National Health Service (NHS) prescribing services,[10] and hospital admission data collected by all English Clinical Commissioning Groups (CCGs decide what services

are needed for diverse local populations, and ensure that they are provided (https://www.england.nhs.uk/ccgs/). They also hold responsibility for local antimicrobial prescribing guidelines. Lessons learnt from the feasibility study also suggested better use of the tool would be facilitated if the intervention was embedded within the practice electronic health record system. The intervention in this study has thus been embedded in the Egton Medical Information Systems (EMIS) electronic patient record system, used in 56% of the primary care practices in England.[11]

### Aims and objectives
The aim of the CHICO RCT is to reduce antibiotic prescribing among children presenting with acute cough and RTI without increasing hospital admission for this condition.

The objectives are to determine whether the CHICO intervention decrease the number of dispensed prescriptions for oral amoxicillin and macrolide antibiotics (the predominant antibiotics given to children presenting with acute cough and RTIs in the UK) for children aged 0–9 years (efficacy comparison) and to determine if the CHICO intervention does not increase hospital admissions for children with a hospital diagnosis of RTI (non-inferiority comparison).

## METHODS AND ANALYSIS
### Study design
The CHICO RCT is an efficient, pragmatic open-label, two-arm (intervention vs usual care) trial with an embedded qualitative study, aimed at reducing antibiotic prescribing among children presenting with acute cough and RTI, with randomisation at the practice level, using routine antibiotic dispensing and hospitalisation data to assess effectiveness.

### Study population, setting and recruitment plan
The study population is children aged 0–9 years presenting with acute cough and RTI. Oral suspensions are more often given to this age group. The setting is consultations in primary care practices with prescribing clinicians in diverse regions across England. Recruitment is at the practice level, so consent is not required for individual participants. Recruitment of practices is via CCGs and by using the Clinical Research Network (CRN) who support patients, the public and health and care organisations across England to participate in high-quality research (https://www.nihr.ac.uk/explore-nihr/support/clinical-research-network.htm). All CCGs are already committed to national AMR strategies and an initial approach to several CCGs about collaboration in this study has been enthusiastically welcomed. CCGs with 15 or more EMIS practices will be targeted and we will use a member of the CCG medicines management team as the primary contact given the established links they already have helping to provide routine data.

## Eligibility

### Inclusion

GP practices in England using the EMIS electronic patient record system where the local CCG has agreed to provide data and the practice consented to take part.

### Exclusion

Practices will be asked directly whether they are participating in any antimicrobial stewardship activities during our study period and these will be recorded. If these activities involve concurrent intervention studies where there is potential to confound or modify the effects of the intervention these practices will be excluded. Practices involved in the CHICO feasibility study or are merging or planning to merge with another practice will also be excluded.

## Treatment arms

### Intervention

The theory-informed intervention[12] consists of both a clinician-focused algorithm to predict risk of hospitalisation for RTI in the following 30 days, in children with acute cough and RTI, and carer-focused personalised information recording decisions made at the consultation and safety netting information.[13]

The algorithm contains seven predictors (mnemonic STARWAVe): Short illness duration (parent/carer reported ≤3 days); raised Temperature (parent/carer reported severe in previous 24 hours or ≥37.8°C on examination); age of child (<2 years); intercostal or subcostal Recession on examination; Wheeze during chest stethoscope examination; history of Asthma and Vomiting (parent/carer reported moderate or severe in the 24 hours prior to consultation). The actions related to the algorithm scores are shown in table 1, in each case the algorithm result (eg, low-risk group) automatically appears and the pop-up text is available if the clinician hovers over the result. The algorithm is intended as a supportive additional component of a consultation in which it is likely that a number of aspects will inform the clinical decision making, including whether or not to prescribe antibiotics.

We will enrol a champion (eg, a GP, nurse or practice manager) at each practice to help encourage and monitor the use of the intervention. These champions will help to set up the intervention and run monthly queries of intervention use via EMIS that will be monitored centrally by the study team.

### Training for practitioners in the intervention arm

The intervention clinicians will be provided with print and on-line evidence-based information to describe why, how and when to use the intervention. A practice champion will distribute the self-directed training materials within the practice and encourage all clinicians to use the intervention appropriately. In the training package for clinicians, it will be emphasised that the primary purpose of the intervention is to support the care of the larger proportion of children (69%) who have a very low risk of hospitalisation.

### Usual care

The clinicians in practices randomised to the comparator arm will be asked to treat children presenting with acute cough and RTI as they would normally. Baseline and follow-up data on control practices will be collected but no data are being collected directly from the clinicians, no practice champions identified or specific contact being made.

## Patient and public involvement

This intervention has been developed collaboratively with our parent advisory group (PAG) and clinical advisory group throughout the 'TARGET' programme. Their comments and suggestions about the format of the intervention and parent/carer materials have informed both the intervention and the design of the earlier feasibility study.

Similar involvement will be sought for the trial. We will seek agreement from a newly formed PAG to meet throughout the study to report on progress of the study and discuss issues that arise during the study. PAG members will input into all the materials for parents/carers as they are further developed including any patient-facing tools. We will also form a clinician and pharmacist advisory group to assist with the implementation and any further refinements to the intervention. They will meet

| Table 1 | Text for algorithm result |
| --- | --- |
| **Algorithm result** | **Pop-up text** |
| Very low-risk group | Very reassuring CHICO score: 0 or 1 CHICO predictors:>99.6% of children will recover from this illness with home care. Consider a no or delayed antibiotic prescribing strategy. CHICO leaflet and letter covers common concerns and safety netting advice. |
| Average risk group | Reassuring CHICO score: 2 or 3 CHICO predictors:>98% of children will recover from this illness with home care. Consider no or delayed antibiotic prescribing strategy. CHICO leaflet and letter covers common concerns and safety netting advice. |
| Elevated risk group | Safety netting needed: 4+CHICO predictors: This is more than average, but >87% of children will still recover from this illness with home care. Highlight SAFETY NETTING advice in CHICO leaflet. |

CHICO, CHIldren's COugh.

once in person and then contribute by Skype or email to refine GP information and intervention delivery.

### Data collection and randomisation

Data collection takes place when both the individual practice and allied CCG agree to participate. Data will be entered onto a purpose designed database, validation and cleaning will be carried out throughout the trial. Only the administrative team and analysts will be able to access this data.

The number of dispensed amoxicillin and macrolides antibiotics given to children aged 0–9 years will be taken from the routine data source, epact2,[10] which provides practice-specific information by each 5-year age epoch. Data will be collected from CCGs for every participating practice with regards to the number of hospitalisations and emergency department attendances for RTIs. Only fully anonymised data sets will be sent from the GP practices and CCGs. This will be sent to a secure NHS e-mail address. We will collect data for the 12-month period each practice will be in the study and the 12-month period prior to randomisation. An 'implementation period' of around 1 month will allow time for the practices to instal the intervention and encourage staff to use it. Any data collected during this period will not be used in the analysis. Where data are suppressed, owing to a low number of events, practices will be asked to provide aggregate 12-month data for baseline and follow-up. Practice list size data, per month and 5-year epoch, will be obtained from the NHS digital website. In the unlikely event that a practice no longer wishes to participate, we will request all outstanding data collected up until the point of withdrawal. For intervention practices only, monthly intervention usage data will be captured. The data will be extracted from the EMIS system and will include how often the intervention is being used and by whom. Fidelity will be measured from the analysis of intervention data usage, scrutiny of the follow-up questionnaires and qualitative interviews.

The trial is supported by the Bristol Randomised Trials Collaboration (BRTC). The trial will conform to the BRTC standard operating procedures. The BRTC central research team will help prepare the trial documentation and data collection forms, specify the randomisation scheme, develop and maintain the study database, check data quality, monitor recruitment and carry out analyses in collaboration with the investigators. Both an independent trials steering committee (TSC) and data monitoring committee (DMC) will be appointed.

### Baseline measurements

All GP practices recruited will be asked to complete a baseline questionnaire prior to randomisation to allow capture of practice characteristics. This includes: (i1) practice staff composition (GP partners/salaried/ sessional nurse practitioners and practice nurses and locums used in the last 12 months); (2) available characteristics (such as postcode, total patients registered); (3) registered child patients—number, age group, ethnicity and gender; (4) triage systems used to handle children presenting with acute cough and RTI and (5) which clinicians prescribe antibiotics to children aged 0–9 years.

### Randomisation

GP practices will be randomised on a 1:1 basis by the independent BRTC. Randomisation of practices will be stratified by CCG, with further minimisation by practice list size and baseline dispensing rates of 0–9 year olds; calculated using data from the 12 months prior to the CCG joining the CHICO study. A trial schematic is shown in figure 1.

### Follow-up measurements

A follow-up questionnaire will be sent to all practices after 12 months (similar to the baseline questionnaire) asking about staffing levels and management of RTI among children as well as use of intervention for those in the intervention arm. Questions will also be included about whether the practice has merged or split with another practice, if they have had any related fatalities in children aged 0–9 years during the 12 months participation and for intervention practices only, their experience of using the intervention, problems encountered and whether they would use it again.

### Blinding

As this is a cluster RCT and due to the nature of the intervention delivery, it will not be possible to blind the practices to their allocation of either control or intervention group. Administrative staff will have access to individual data items, for entry into the database. The statistician will have access to aggregate information, by arm, to be able to report to the DMC and monitor hospitalisations.

### Outcomes

The primary and secondary outcomes are listed in table 2. All practices will collect data over a 12-month period, thus any seasonal fluctuations will be captured.

### Safety reporting

Adverse events (AE) and serious AEs (SAE) will be recorded and reported in accordance with Good Clinical Practice guidelines and the Sponsor's Research Related Adverse Event Reporting Policy. This trial is a low risk study, SAEs will only be reported if they are fatal or serious and potentially related to trial participation (ie, they result from advice provided by the intervention algorithm). As one of the outcomes for the trial is hospitalisation, we do expect some participants to be admitted to hospital (due to a deterioration of their underlying illness). Hospitalisation due to RTI is an expected SAE and will not be subject to expedited reporting. Both SAEs and hospitalisation rates will be regularly reported to the DMC who will raise any safety concerns to the trial team and TSC for further action. Expected SAEs include but are not limited to pneumonia, empyema, deteriorating bronchiolitis.

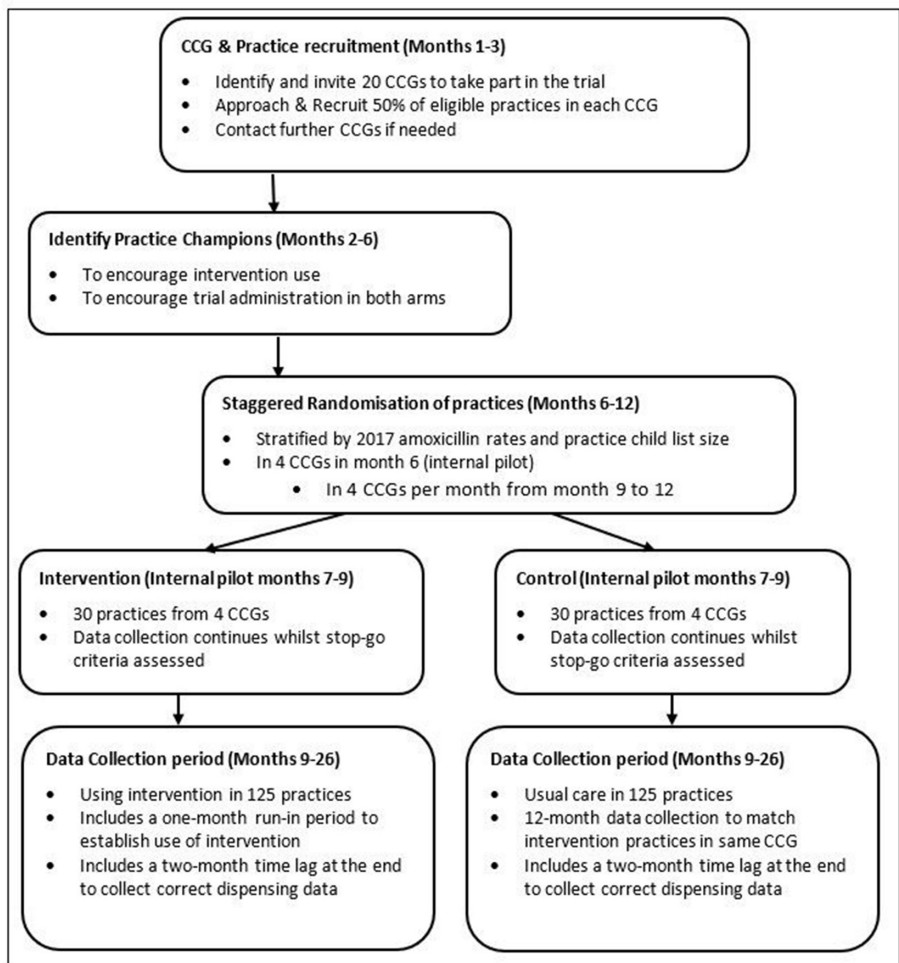

**Figure 1** Trial schematic. CCGs, Clinical Commissioning Groups.

### SAEs related to the use of intervention

If the GP practice champion or attending clinician suspects that an SAE resulted from use of the intervention it should be reported to the study team immediately. The causality of the event will be assessed by the practice clinician and a delegated clinician working within the CHICO study team. If the event is deemed to be probably or definitely related to the intervention the SAE will be reported to the Research Ethics Committee and sponsor according to the expedited timescales.

### Fatal SAEs

All practices should inform the study team immediately of any fatal SAEs in children that had presented with RTI at a practice consultation and were 0–9 years old at the time of consultation. This applies to any deaths occurring within 90 days of the consultation.

### Internal pilot study

An internal pilot phase lasting 3 months and using 4 or more CCGs to recruit 60 practices will help establish how many CCGs we will eventually need to approach. Stop-go (traffic light) criteria will be used for (1) practice recruitment, (2) identification of a practice

champion, (3) intervention use and (4) ability to obtain dispensing data from the CCGs. A green light will be given for 80+% success (90% for dispensing data) and an amber light to implement remedial action at 70%–79% (80%–89% for dispensing data). A red light would indicate either a further pilot is needed or stopping the trial.

### Sample size determination

Both sample size calculations assume 90% power and a conservative two-sided alpha of 0.025 to take account of the two coprimary outcomes. Both sample sizes also assume an intracluster correlation coefficient of 0.03 (which has been described as the upper CI for ICCs in efficient cluster randomised trials,[14][15] an estimated coefficient of variation of 0.65 (to take account of differences in cluster size[16] and an assumption of 750 children on average aged 0–9 years registered per practice (based on Bristol & Bath CCG data). Expected differences assumed: (1) a reduction in dispensing rate from 33 prescriptions per 100 registered children aged 0–9 years to 29 (or fewer) prescriptions (ie, ≥10% overall reduction); and (2) a hospitalisation rate that is no more than 2% in the intervention arm, compared with the control arm which

**Table 2**  Detailed study outcomes

| Primary outcomes | |
|---|---|
| P1) | Whether the CHICO intervention decrease the number of dispensed prescriptions for oral amoxicillin and macrolide antibiotics* (efficacy comparison). |
| P2) | Whether the CHICO intervention result in no increase in hospital admissions† for children with a hospital diagnosis of RTI (non-inferiority comparison). |
| **Secondary Outcomes** | |
| S1) | Whether the CHICO intervention results in no change in the emergency department attendance rates‡ of children with a diagnosis of RTI. |
| S2) | The costs to the NHS of using the CHICO intervention (health economic outcome). |
| S3) | Whether there is any intervention effect modified by the no of locums used in the practice (treatment interaction). |
| S4) | Whether there is any intervention effect modified by the practices' prior antibiotic prescribing rate (treatment interaction). |
| S5) | Whether the effects of the CHICO intervention differ between practices with or without nurse prescribers (treatment interaction).§ |
| S6)¶ | Whether the effects of the CHICO intervention differ between practices with one site vs multiple sites (branches) at each practice (treatment interaction). |
| S7)¶ | Whether the effects of the CHICO intervention differ between practices with follow-up prior to COVID-19 pandemic and during the COVID-19 pandemic (treatment interaction). |
| S8)¶ | Whether the effects of the CHICO intervention differ in areas of high/low deprivation. |
| S9) | Whether the effects of the CHICO intervention differ within child age groups. |
| S10) | Whether the use of the CHICO intervention varies between practices (adherence) and over time (seasonal differences) and the influence this has on the dispensing rates. |
| S11) | Whether the embedded CHICO intervention is acceptable to, and used by, primary care clinicians (GPs and practice nurses). |

*The dispensing rate, calculated by adding the number of amoxicillin and macrolide antibiotics dispensed over the follow-up year divided by the number of children aged 0–9 years (median monthly list size) at each practice over the 12-month follow-up period.
†The rate of hospital admission for RTI among children aged 0–9 years using the same denominator as above.
‡This is a secondary outcome already collected from practices by CCGs.
§If a large majority of practices have nurse prescribers then we may look at this as a continuous percentage of nurse prescribers, out of all GP and nurse prescribers.
¶Added after the trial began, due to unforeseen circumstances including more variability in practices than we first anticipated. Therefore, these do not match those listed in the trial registration.
CCGs, Clinical Commissioning Groups; CHICO, CHICO; GP, general practitioner; NHS, National Health Service; RTI, respiratory tract infection.

is estimated to be 1%. This is based on a non-inferiority margin of 1%, however, the investigators wanted to err on the side of caution and use a two-sided alpha for the sample size calculation. This gave an overall sample size requirement of 310 practices; 155 intervention and 155 control practices.

### Economic evaluation
To address our secondary aims (S2) a focus on costs will clarify whether and by how much NHS costs might change in the event of a widespread deployment of the algorithm into routine clinical practice. Given the light-touch efficient design of the trial, the economic evaluation will be limited to a between-arm comparison of mean NHS costs in a cost–consequence analysis. NHS costs will be calculated from the costs of the intervention itself, prescriptions of amoxicillin and macrolides per the coprimary outcome, ED attendances and hospital admissions.

### Qualitative study
Qualitative interviews with clinicians (GPs and practice nurses) and other practice staff (managers, pharmacists) and CCG staff (medicines managers) will explore the use of the intervention, how it was embedded into practice and whether it was used appropriately. The interview topic guide will be informed by normalisation process theory (NPT) developed to explain the social processes leading to routine embedding of complex interventions in healthcare.[17 18]

NPT proposes that implementation of interventions is dependent on the ability of participants to fulfil four criteria; 'coherence' (how people make sense of the intervention), 'cognitive participation' (the work to develop new practices), 'collective action' (the work to operationalise practices) and 'reflexive monitoring' (ways in which people appraise how new practices are working).

Clinicians and other key staff from the intervention practices will be invited to participate in semistructured

interviews to explore their views and experiences of the intervention. Audio recorded verbal consent will be taken from participants. The first set of interviews will be conducted during the internal pilot phase and findings fed back to help guide best practice during the rest of the study. A second phase of interviews will be conducted when the clinicians have been using the intervention for several months to investigate the normalisation and sustainability of using the intervention. Interviews are expected to take 30–45 min.

Purposive sampling will be used to include a maximum variation sample to take account of: clinical experience, dispensing rates of practices and practices serving areas of high and low socialeconomic deprivation. The sample sizes will be determined by the need to achieve data saturation, such that no new themes are emerging from the data by the end of data collection.[19] Interviews will be analysed in batches. This is likely to include up to 30 clinicians and 20 other staff involved in implementation.

## Data analysis
### Quantitative data analysis
All analyses and reporting will be in line with Consolidated Standards of Reporting Trials (CONSORT) guidelines and its extension for cluster randomised trials.[20] Primary analyses will be conducted on an intention-to-treat basis, a per-protocol analysis will also be conducted as part of the sensitivity analyses. A full CHICO statistical analysis plan will be developed and agreed by the TSC prior to undertaking analyses of the main trial. The statistical analysis plan will include health economics and qualitative analysis subsections. At the end of the trial, all outcomes will be described and compared with the appropriate descriptive statistics where relevant: mean and SD for continuous and count outcomes, medians and IQR if required for skewed data and numbers and percentages for dichotomous and categorical outcomes. Depending on the dispersion of the data we may use linear regression or a random effects Poisson regression (negative binomial regression) model to analyse both coprimaries, with CCG included as a random effect. This has the advantage of incorporating person-years follow-up (number of children at a practice multiplied by the length of follow-up for that practice) and examining clustering by CCG. Each coprimary will be adjusted for baseline dispensing rates or hospitalisation rates, using the 12 months of data collected prior to randomisation. Effects of number of practices within CCGs and number of patients within each practice will also be investigated in a sensitivity analysis. Other baseline characteristics between practices will be examined to ensure randomisation is balanced in the two arms. Any differences in excess of 0.5 SDs or 10% or more will be controlled for in sensitivity analyses to ensure that the imbalance does not affect the overall result. The effects of missing data will be explored using sensitivity analyses. We anticipate no more than 10% missing data and that it will be missing at random. The pattern and extent of missing data will be

explored and any changes to the methods described in the analysis plan will be fully justified in the study report and publication. All quantitative data will be analysed using Stata version 15.

### Qualitative data analysis
Interviews will be transcribed and anonymised. Analysis will inform further data collection, for instance, analytical insights from data gathered in earlier interviews will help identify any changes that need to be made to the topic guides during later interviews. Qualitative analysis of the transcripts will follow recognised thematic analysis procedures using NVivo software.[21] Thematic analysis,[22] using a data-driven inductive approach,[23] will be used to scrutinise the data in order to identify and analyse patterns and themes of particular salience for participants and across the dataset.[24]

## Study duration and timeline
The initial duration was 33 months from 1 March 2018 to 30 November 2020 although a subsequent extension of 12 months has been awarded to extend the study to 30 November 2021 to recruit the target number of practices. The timeline includes study set up (8 months), internal pilot (3 months), recruitment of practices via CCGs (15 months), follow-up of data collection (12 months) and analysis (7 months).

## Study management
The study will be monitored and audited in accordance with the Sponsor's policy, which is consistent with the UK Policy Framework for Health and Social Care Research. The following data monitor checks will be carried out by the coordination team; that data collected are consistent with adherence to the study protocol; that Case Report Forms (CRFs) are only being completed by authorised persons; that SAE recording and reporting procedures are being followed correctly and that no key data are missing and that data are valid.

## Trial oversight
The study is overseen by a trial management group that meet on a monthly basis and consist of the chief investigator (CI), grant holders, study sponsor and any other staff responsible for the delivery of the trial. The TSC provide independent supervision of the trial and oversees trial progress. The TSC consists of an independent chair (GP and Clinical Academic) and four other independent members including a statistician, a second clinician and two patient and public involvement representative, as well as the CI. The DMC monitors patient safety and trial data efficacy and consists of an independent chair, two other independent members, the CI and trial statistician.

All SAE's are recorded and notified as appropriate to the relevant authorities. The University of Bristol is acting as sponsor for this trial and is responsible for overall oversight of the trial.

## ETHICS AND DISSEMINATION

### Ethics

We are not recruiting individual patients to this study and the primary outcome data are already collected routinely, thus, we do not need patient consent. We will consent the individual practices and encourage all clinicians in the intervention practices to use the intervention tool appropriately. The intervention is directed at the clinician primarily to change their prescribing behaviour. Any data collected from individual clinicians will be anonymised. The personalised letter given to the patients will not contain information on risk of hospitalisation, but rather details of the consultation and the usual safe-guarding information. The CHICO RCT falls under the remit of draft guidance[25] for 'simple and efficient trials' due to the nature of the intervention and the low level of risk involved for patients and meets the suggested principles provided by NHS Health Research Authority.[26]

### Dissemination

A comprehensive plan for disseminating CHICO results will be developed and outputs from this research will comply with the CHICO RCT publication policy and internationally accepted guidelines (CONSORT). The results of the study will be published in the academic press and all GP practices will be offered a lay summary of the main findings of the study. We will disseminate the findings both at a primary care level via CCGs and national conferences as well as international conferences. Whether or not the trial provides evidence of effect we will provide evidence of the potential benefits or pitfalls of an efficiently designed trial; including the utility of routine data collection; the capacity to collect data through current practice systems and the effectiveness of using practice champions and progress feedback to encourage use of such interventions.

### Trial status

Currently (July 2020) 261 EMIS GP practices have been greenlighted across 15 CRN regions in the UK. The first GP practice was recruited to the study in September 2018, with recruitment currently ongoing.

**Author affiliations**

[1]Centre for Academic Primary Care, Bristol Medical School: Population Health Sciences, University of Bristol, Bristol, UK

[2]Bristol Trials Centre (Bristol Randomised Trial Collaboration), Bristol Medical School, University of Bristol, University of Bristol, Bristol, UK

[3]Centre for Academic Child Health, Bristol Medical School, Population Health Sciences, University of Bristol, Bristol, UK

[4]School for Policy Studies, Unversity of Bristol, Bristol, UK

[5]Regional Antimicrobial Stewardship Lead South West Region, NHS Improvement, London, UK

[6]King's College London, London, UK

[7]School of Primary Care Population Sciences and Medical Education, University of Southampton, Southampton, UK

**Acknowledgements** This study was designed and delivered in collaboration with the Bristol Randomised Trials Collaboration (BRTC), part of the Bristol Trials Centre, is in receipt of National Institute for Health Research CTU support funding. The University of Bristol is acting as the sponsor for this trial and the trial is hosted by the NHS Bristol, North Somerset and South Gloucestershire Clinical Commissioning Group (CCG). The authors would like to thank all General practices, CCGs and CRNs for their involvement in CHICO. The authors would also like to thank the members of the TSC and DMC.

**Contributors** ADH, PSB, PJL, NF and JI were responsible for developing the research questions. PSB, ADH, JI, PJL, CCab, CCle, EB, MCG, JH, STC, AL and NF and are responsible for the study design and collection of data. PSB, JT and SB are responsible for study management and coordination. GY, PD and CCle are responsible for the analysis of the data. PSB drafted the paper. All authors read, commented on and approved the final manuscript.

**Funding** This research is funded by the National Institute for Health Research (NIHR) Health Technology Assessment (HTA) programme (funder ref: 16/31/98).

**Disclaimer** The views expressed are those of the authors and not necessarily those of the NIHR or the Department of Health and Social Care.

**Competing interests** None declared.

**Patient and public involvement** Patients and/or the public were involved in the design, or conduct, or reporting, or dissemination plans of this research. Refer to the Methods section for further details.

**Patient consent for publication** Not required.

**Provenance and peer review** Not commissioned; externally peer reviewed.

**ORCID iDs**

Jenny Ingram http://orcid.org/0000-0003-2366-008X
Clare Clement http://orcid.org/0000-0002-5555-433X
Martin C Gulliford http://orcid.org/0000-0003-1898-9075
Nick Francis http://orcid.org/0000-0001-8939-7312
Peter S Blair http://orcid.org/0000-0002-7832-8087

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
