## [Reviewer comments · BMJ Open]

ARTICLE DETAILS

TITLE (PROVISIONAL)	Protocol for an 'efficient design' cluster randomised controlled trial to evaluate a complex intervention to improve antibiotic prescribing for CHildren presenting to primary care with acute COugh and respiratory tract infection: The CHICO study
AUTHORS	Blair, Peter; Seume, Penny; Bevan, Scott; Young, Grace; Ingram, Jenny; Clement, Clare; Cabral, Christie; Lucas, Patricia; Beech, Elizabeth; Taylor, Jodi; Horwood, Jeremy; Dixon, Pdraig; Gulliford, Martin; Francis, Nick; Creavin, Sam T; Lane, Athene; Hay, Alastair

VERSION 1 – REVIEW

REVIEWER	Qiang Sun Shandong University P.R. China
REVIEW RETURNED	23-Aug-2020

GENERAL COMMENTS	In general, it is a well designed trial, the results would contribute to the control AMR if it can spread to other countries with high antibiotic use. For better understanding by international readers, it is suggested to add some background information, such as the roles of CCGs, why choose oral amoxicillin and macrolide antibiotics.
---

REVIEWER	Neil Desai University of British Columbia Dept of Pediatrics and British Columbia Children's Hospital, Canada
REVIEW RETURNED	16-Sep-2020

GENERAL COMMENTS	This is a very interesting pragmatic RCT evaluating the utility of an EMR-embedded clinical decision support tool in reducing antibiotic prescribing for respiratory tract infections in primary care offices throughout England. The use of deidentified data Page 8, line 11: While overall antibiotic prescribing has declined in most settings, the prevalence of inappropriate broad-spectrum prescribing has increased. I mention this to give context to my later comments emphasizing the latter. Page 8, line 18: This is a minor issue, but I might consider removing "the slowing in development of new antibiotics," (or rephrasing), as this is not a cause of antimicrobial resistance. Page 8, line 27: There is an accidental question mark following clinical uncertainty. Page 9, line 11: While overall these are the most common antibiotics prescribed for RTI and cough (especially azithromycin for the latter), the most recent trend in many outpatient settings is a climbing prevalence of broad spectrum prescribing. Please justify why these were not included.
--

	Page 9, line 11: Why was this age group chosen specifically? Please justify. Page 10, line 18: Please clarify why you are expecting the use of an algorithm which, by box 1, predicts risk of admission, to improve antibiotic prescribing? Has it been demonstrated to do this in another setting? My understanding from earlier in your manuscript is that during the feasibility trial there was no significant difference in antibiotic prescribing rate between groups. Page 12, line 34: As demonstrated by Gerber et al. in the references below, the improvement in antibiotic prescribing attributed to an educational intervention was reversed following withdrawal of audit. Similarly, while both groups may demonstrate an improvement during the planned observation period, prolonging the data collection period may reveal the more long term changes in prescribing behavior. JAMA 2013. 309:2345 JAMA 2014. 312:2569
--	---

REVIEWER	Carl Llor University Institute in Primary Care Research Jordi Gol, Via Roma Health Centre, Barcelona.
REVIEW RETURNED	01-Nov-2020

GENERAL COMMENTS	This protocol is clear. I only have some minor comments. In my opinion, if readers are not aware of the previous feasibility study, also published in this journal, they might find it difficult to understand why you use an efficient design for the current study. I would highlight in the introduction section that the overall antibiotic prescribing rates for respiratory tract infections were significantly higher for the intervention group compared to the control group in your previous trial, mainly because of the different severity of the infections recruited in the two different arms. How would you cope with the covid19 pandemic? We all know about the organizational changes suffered in primary care due to this pandemic and how this can hinder the recruitment of patients. You are planning to end this study by November 2021, but this could be delayed. Please discuss. You are recruiting patients with acute cough and RTI. Could you be more explicit regarding the inclusion criteria? Are you also recruiting children with upper RTIs like sore throat (they can also present with acute cough), acute otitis media, etc? You are recruiting research-naïve practices – practices not having participated in the feasibility trial, not participating in any antimicrobial stewardship activities, etc -. This can be challenging now, as most of them might have been participated in similar studies. Please discuss. Why can't you collect information on delayed antibiotic prescribing in the two groups? Don't you consider that the percentage of this can be different in the intervention group compared to the control?
--

VERSION 1 – AUTHOR RESPONSE

Reviewer: 1

Comments to the Author

In general, it is a well designed trial, the results would contribute to the control AMR if it can spread to

other countries with high antibiotic use. For better understanding by international readers, it is suggested to add some background information, such as the roles of CCGs, why choose oral amoxicillin and macrolide antibiotics.

Thank you for your comments. We have added an explanatory sentence about CCGs on page 4 'English Clinical Commissioning Groups (CCGs decide what services are needed for diverse local populations, and ensure that they are provided (<https://www.england.nhs.uk/ccgs/>)..They also hold responsibility for local antimicrobial prescribing guidelines'

CRNs on page 5

'Recruitment of practices is via CCGs and by using the Clinical Research Network (CRN) who support patients, the public and health and care organisations across England to participate in high-quality research (<https://www.nihr.ac.uk/explore-nihr/support/clinical-research-network.htm>) .'

And explained why we are measuring amoxicillin and macrolide use on page 5

'...oral amoxicillin and macrolide antibiotics (the predominant antibiotics given to children presenting with acute cough and respiratory tract infections in the UK) for children aged 0-9 years'

Reviewer: 2

Comments to the Author

This is a very interesting pragmatic RCT evaluating the utility of an EMR-embedded clinical decision support tool in reducing antibiotic prescribing for respiratory tract infections in primary care offices throughout England. The use of deidentified data

Thank you

Page 8, line 11: While overall antibiotic prescribing has declined in most settings, the prevalence of inappropriate broad-spectrum prescribing has increased. I mention this to give context to my later comments emphasizing the latter.

Ok, see our later response.

Page 8, line 18: This is a minor issue, but I might consider removing "the slowing in development of new antibiotics," (or rephrasing), as this is not a cause of antimicrobial resistance.

Agreed. This has now been removed

Page 8, line 27: There is an accidental question mark following clinical uncertainty.

Thank you. This has now been removed.

Page 9, line 11: While overall these are the most common antibiotics prescribed for RTI and cough (especially azithromycin for the latter), the most recent trend in many outpatient settings is a climbing prevalence of broad spectrum prescribing. Please justify why these were not included.

We agree that there are concerns regarding broad spectrum antibiotic prescribing in primary care, but this is mainly used among adults and the NHS Improvement AMR policy has led to a 40% reduction in broad spectrum prescribing in the last 5 years according to the ESPAUR report (<https://www.gov.uk/government/publications/english-surveillance-programme-antimicrobial-utilisation-and-resistance-espaur-report>). The most common antibiotics prescribed for children with acute cough and respiratory infection are amoxicillin and, where penicillin-allergic, macrolides (<https://www.nice.org.uk/guidance/ng120/resources/visual-summary-pdf-6664861405>).

Page 9, line 11: Why was this age group chosen specifically? Please justify.

The overall aim of our 5-year programme grant (TARGET) was to improve the use of antibiotics for children with respiratory infections. One of the TARGET workstreams was charged with understanding the reasons for high antibiotic prescribing in this group. Qualitative work established that clinicians regard children with respiratory infections as vulnerable to subsequent (post consultation) deterioration, and that clinicians were particularly concerned not to be seen to have under-treated a child who is

subsequently admitted to hospital. Thus, we found 'just-in-case' antibiotic prescribing was common because clinicians were unable to predict which children would/would not be admitted in the days following primary care-assessment. In TARGET we collected data on over 8,000 children aged 3 months to 16 years presenting with acute cough and respiratory tract infection. The vast majority of these children (~90%) were less than 10 years old. Also CCG data on antibiotic prescribing are collected in 5 year epochs (0-4 years, 5-9 years etc). These are pragmatic reasons, but the main reason was that oral suspensions (which we are trying to capture) tend to be given to younger children. We therefore have added a sentence on page 5.

"The study population is children aged 0-9 years presenting with acute cough and RTI. Oral suspensions are more often given to this age group."

Page 10, line 18: Please clarify why you are expecting the use of an algorithm which, by box 1, predicts risk of admission, to improve antibiotic prescribing? Has it been demonstrated to do this in another setting? My understanding from earlier in your manuscript is that during the feasibility trial there was no significant difference in antibiotic prescribing rate between groups.

The feasibility study does not test the efficacy of the intervention but whether it is possible to do a trial. The direction of the results of the feasibility study showed that antibiotic prescribing levels in both arms was significantly reduced compared to the data from our previous cohort study and even lower in the control arm. In our publication of these findings we produced evidence of differential recruitment where clinicians (especially in the control arm) were recruiting healthier children. We have now added more clarity on page 4.

'The subsequent feasibility cluster randomised controlled trial [RCT] for CHildren's COugh (CHICO) showed significant prescribing reductions in both arms of the trial compared to the cohort data of the programme, but also exposed both lower prescribing levels and differential recruitment of healthier children in the control arm.'

Page 12, line 34: As demonstrated by Gerber et al. in the references below, the improvement in antibiotic prescribing attributed to an educational intervention was reversed following withdrawal of audit. Similarly, while both groups may demonstrate an improvement during the planned observation period, prolonging the data collection period may reveal the more long term changes in prescribing behavior.

JAMA 2013. 309:2345

JAMA 2014. 312:2569

We agree that there can be a change of behaviour among clinicians and practices under trial conditions especially if they are heavily involved in the trial process and feel under scrutiny. This is precisely why we have developed a more 'hands-off' efficient design approach that excludes individual recruitment, does not resort to visiting practices to search patient notes and uses reliable data already collected independently at the practice level as one of the primary outcomes. Indeed using this approach means that longer term impact can be monitored both with the trial groups and beyond if this intervention is rolled out, as the primary outcome will continue to be collected after our trial ends.

Reviewer: 3

Comments to the Author

This protocol is clear. I only have some minor comments. In my opinion, if readers are not aware of the previous feasibility study, also published in this journal, they might find it difficult to understand why you use an efficient design for the current study. I would highlight in the introduction section that the overall antibiotic prescribing rates for respiratory tract infections were significantly higher for the intervention group compared to the control group in your previous trial, mainly because of the different severity of the infections recruited in the two different arms.

We have now tried to make this clear (please see response to reviewer 2 comments)

How would you cope with the covid19 pandemic? We all know about the organizational changes suffered

in primary care due to this pandemic and how this can hinder the recruitment of patients. You are planning to end this study by November 2021, but this could be delayed. Please discuss.

When we began writing this manuscript the pandemic had just begun so the impact on the study was not clear. Subsequently we have had to delay recruitment for 4 months (and now have a 4 month non-costed extension to March 2022) and have contacted the intervention practices to find out how consultations and the use of the intervention have changed. Thus it might be better to discuss this impact in a later paper rather than here but we are happy to reconsider if the editors don't mind extending the word limit of this manuscript. If so we would add the description below as an additional paragraph:

"The COVID pandemic has had some impact on the trial. When the number of cases began to rise in England in March 2020 we halted recruitment, stopped reminders for intervention usage and follow-up data to alleviate any administrative burden and advised practices that they could stop using the intervention if they so wished. Practice recruitment began again in July 2020 and a 4 month un-costed extension was granted giving a new study end-date of March 2022. An on-line survey of intervention practices suggested that consultations between March and July 2020 were a mixture of face-to face, video and telephone contacts and some clinicians preferred to continue with the intervention during this period using workarounds such as asking the carer to take the temperature of the child. Given we are conducting an intention to treat analysis, requiring 12 calendar months data collection from each practice and can collect the primary outcome data despite these interruptions we will continue with the statistical analysis as planned. We will use the qualitative interviews with clinicians to capture more detail of organisational changes in primary care and specific changes in consultations that may affect intervention use to inform our sensitivity analyses. We can retrospectively measure intervention use (as this is embedded and can be monitored via the EMIS system) and sensitivity analyses will include investigating differences pre-COVID, during the pandemic and (hopefully) post-COVID".

You are recruiting patients with acute cough and RTI. Could you be more explicit regarding the inclusion criteria? Are you also recruiting children with upper RTIs like sore throat (they can also present with acute cough), acute otitis media, etc?

We are not recruiting patients but rather practices and advise intervention clinicians to use the tool for children aged 0-9 years presenting with acute cough and respiratory tract infections. We do not give a prescriptive list of exclusions. We agree that acute cough is a symptom present in both upper and lower respiratory tract infection. However, it is more commonly (than sore throat and otitis media) associated with lower respiratory infection and the subsequent development of respiratory distress/ pneumonia requiring hospital admission. It is also the most common reason for primary healthcare attendance world-wide [Okkes M, SK O, H L. The Probability of Specific Diagnoses for Patients Presenting with Common Symptoms to Dutch Family Physicians. *Journal of Family Practice* 2002;51:31-36]. As recommended for research in primary care by the Medical Research Council, [Medical Research Council. Primary Health Care Research Review. MRC Topic Review. London, 1997] our inclusion criteria were designed to reflect routine primary care practice, as part of which clinicians will often not be sure if a respiratory infection is predominantly upper, lower or both.

You are recruiting research-naïve practices – practices not having participated in the feasibility trial, not participating in any antimicrobial stewardship activities, etc -. This can be challenging now, as most of them might have been participated in similar studies. Please discuss.

There are nearly 8,000 GP practices in England and even the largest studies only recruit a few 100 at a time. We asked practices to opt out if they were in a conflicting study, the main other study at the time being the ARCTIC PC study. Of the 457 expressions of interest received, only 8 were excluded due to having been in the feasibility study and 11 due to taking part in ARCTIC PC. The latter were subsequently recruited once they had ended participation in that study. We have recruited nationally and the study now has GP practices in all 15 CRNs in England. Compared to other studies we impose less burden on clinicians (as they don't have to recruit patients) and practices (as we don't have to search for patient notes or ask anything of the control practices other than advertise to their patients that the study is being conducted). We see this as an advantage of our efficient design and will report in a future paper of our baseline results how many practices were research-naïve and any difficulties we have had

with recruitment. The protocol paper is probably not the place to discuss this although we thank the reviewer for this observation and will flag this up by adding a sentence on page 4.

'This simpler design, placing fewer demands on clinicians and practices compared to other studies, will also encourage the recruitment of research-naïve practices.'

Why can't you collect information on delayed antibiotic prescribing in the two groups? Don't you consider that the percentage of this can be different in the intervention group compared to the control?

We agree that the use of delayed prescribing is important and that it would have been interesting to examine the effects of our intervention on this strategy. However, delayed prescribing is rarely coded in primary care, so establishing which prescriptions were delayed is not possible. However, our outcome is dispensed (vs. prescribed) antibiotics, as described on page 4, and so represents an overall measure of both 'immediate' and 'delayed' prescribing.

VERSION 2 – REVIEW

REVIEWER	Qiang Sun Shandong University, China
REVIEW RETURNED	27-Dec-2020

GENERAL COMMENTS	A good protocol aimed to assess whether embedding a multi-faceted intervention into GP practice IT systems will result in reductions of antibiotic prescribing without impacting on hospital attendance for RTI. No other comments.
--

REVIEWER	Neil Desai University of British Columbia, BC Children's Hospital Canada
REVIEW RETURNED	13-Jan-2021

GENERAL COMMENTS	This is a very valuable and interesting study, investigating the utility of a clinical care decision tool in reducing prescribing for acute respiratory tract infections in the community.
--

REVIEWER	Carl Llor University research institute Jordi Gol, Barcelona
REVIEW RETURNED	25-Dec-2020

GENERAL COMMENTS	This is a well designed trial. The protocol is now more understandable.
---